# The M2a Macrophage Phenotype Accompanies Pulmonary Granuloma Resolution in *Mmp12* Knock-Out Mice Instilled with Multiwall Carbon Nanotubes

**DOI:** 10.3390/ijms222011019

**Published:** 2021-10-13

**Authors:** David Ogburn, Sophia Bhalla, Nan Leffler, Arjun Mohan, Anagha Malur, Achut G. Malur, Matthew McPeek, Barbara P. Barna, Mary Jane Thomassen

**Affiliations:** 1Division of Pulmonary, Critical Care and Sleep Medicine, Department of Internal Medicine, Brody School of Medicine Sciences Building, East Carolina University, Moye Boulevard, Greenville, NC 27834, USA; davidogburn4@gmail.com (D.O.); bhallas21@ecu.edu (S.B.); lefflern@ecu.edu (N.L.); malura@ecu.edu (A.M.); mmcpeek@email.unc.edu (M.M.); mutchka@oberlin.net (B.P.B.); 2Division of Pulmonary Disease and Critical Care Medicine, Department of Internal Medicine, Virginia Commonwealth University, Richmond, VA 23284, USA; arjun.mohan@vcuhealth.org; 3Department of Microbiology & Immunology, St. George’s University, St. George 999166, Grenada; amalur@sgu.edu

**Keywords:** sarcoidosis, MMP12, PPARγ, MWCNT, granuloma, inflammation

## Abstract

Sarcoidosis is a chronic disease with unknown etiology and pathophysiology, characterized by granuloma formation. Matrix Metalloproteinase-12 (MMP12) is an elastase implicated in active granulomatous sarcoidosis. Previously, we reported that oropharyngeal instillation of multiwall carbon nanotubes (MWCNT) into C57Bl/6 mice induced sarcoid-like granulomas and upregulation of MMP12. When *Mmp12* knock-out (KO) mice were instilled with MWCNT, granuloma formation occurred 10 days post-instillation but subsequently resolved at 60 days. Thus, we concluded that MMP12 was essential to granuloma persistence. The aim of the current study was to identify potential mechanisms of granuloma resolution in *Mmp12*KO mice. Strikingly, an M2 macrophage phenotype was present in *Mmp12*KO but not in C57Bl/6 mice. Between 10 and 60 days, macrophage populations in MWCNT-instilled *Mmp12*KO mice demonstrated an M2c to M2a phenotypic shift, with elevations in levels of IL-13, an M2 subtype-regulating factor. Furthermore, the M2 inducer, Apolipoprotein E (ApoE), and Matrix Metalloproteinase-14 (MMP14), a promoter of collagen degradation, were upregulated in 60-day MWCNT-instilled *Mmp12*KO mice. In conclusion, alveolar macrophages express two M2 phenotypes in *Mmp12*KO mice: M2c at 10 days when granulomas form, and M2a at 60 days when granulomas are resolving. Findings suggest that granuloma resolution in 60-day *Mmp12*KO mice requires an M2a macrophage phenotype.

## 1. Introduction

Sarcoidosis is a multisystem granulomatous disease [1]. There are approximately 185,000 sarcoidosis cases in the United States, and approximately 1.2 million worldwide [1,2,3]. The pathogenesis of sarcoidosis is complex and not currently understood [1,4], and the current use of glucocorticoids as the primary treatment has shown mixed results along with severe adverse effects [5]. Sarcoid granulomas typically present in the eyes, skin, and lungs, with pulmonary granulomatous formations being associated with increased disease severity [6]. We previously described a multiwall carbon nanotube (MWCNT)-based murine model of pulmonary granulomatous inflammation in C57Bl/6 wild-type mice [7]. This model has been shown to induce chronic pulmonary granulomas with a histological presentation resembling that of human granulomas [7].

Alveolar macrophages, the predominant immune cell in the airways of healthy individuals, have been identified as key players in sarcoidosis pathogenesis [8]. Peroxisome proliferator-activated receptor gamma (PPARγ) is a transcription factor that negatively regulates pro-inflammatory macrophage activation [9]. PPARγ is highly expressed in healthy individuals; however, its expression is decreased in alveolar macrophages from human sarcoidosis patients [10,11]. Alveolar macrophages from MWCNT-instilled mice also exhibit decreased PPARγ expression relative to control [12]. Correspondingly, interferon gamma (IFNγ), a pro-inflammatory cytokine, is upregulated in both human sarcoid patients and in the MWCNT model [12,13].

MMPs are a class of proteases that primarily function to degrade the extracellular matrix [14]. Additionally, this family of proteins functions in the immune and inflammatory responses, and are involved in tissue repair, tissue remodeling, and cell proliferation [14]. A transcriptional survey comparing alveolar macrophages from human sarcoid patients to alveolar macrophages from the MWCNT model found that a protein in this family, namely MMP12, was the most highly expressed gene in both groups [13]. MMP12 is an elastase produced by alveolar macrophages [14,15], and has previously been implicated in many chronic inflammatory diseases, both pulmonary and systemic [14]. In sarcoidosis patients, MMP12 gene expression is high in areas of active granulomatous inflammation [16]. Based on these observations, we analyzed pulmonary responses of *Mmp12* KO mice to MWCNT instillation.

Previous results indicated that MWCNT instillation of *Mmp12* KO mice resulted in granuloma formation at 10 days [17]. However, in a time course comparison, it was observed that while granulomas persist through 60 days in C57Bl/6 mice, granulomas in *Mmp12* KO mice were resolving [17]. These results suggested that MMP12 was required for granuloma persistence. In MWCNT-instilled C57Bl/6 mice, PPARγ gene expression is decreased and IFNγ gene expression is elevated at 60 days [12]. In contrast, when *Mmp12* KO mice are instilled with MWCNT, PPARγ expression is increased and IFNγ expression is decreased at 60 days [17].

PPARγ expression coincides with granuloma resolution in the *Mmp12* KO model. In addition to the regulation of cytokines in the immune response, PPARγ can alter the phenotype of alveolar macrophages [18]. PPARγ promotes the M2 macrophage phenotype [19,20]. M2 macrophages function to heal, with anti-inflammatory activity in tissue repair and in the maintenance of tissue integrity [21,22]. In contrast, the M1 macrophages are pro-inflammatory in nature with their primary function being to kill or clear foreign antigens or pathogens by phagocytosis or microbicidal activity [21]. Interestingly, ApoE, a glycoprotein that may be induced by PPARγ through LXR activation [23,24], also induces an anti-inflammatory phenotype in macrophages, which drives production of factors such as IL-13 that promote the alternative M2 phenotype activation [25]. Based on these observations, we hypothesized that ApoE might also play a role in transforming the *Mmp12* KO macrophage phenotype into a form responsible for granuloma resolution.

## 2. Results

### 2.1. Alveolar Macrophages from Mmp12 KO Mice Express M2 Marker CD 206 (MRC1)

Previous and current studies have demonstrated prominent granuloma formation at 10 days post-MWCNT instillation in both C57Bl/6 wild-type and *Mmp12* KO mice [17]. However, at 60 days, *Mmp12* KO mice revealed markedly attenuated granuloma formation together with elevated PPARγ expression as compared to wild-type [17] (Figure 1A). Since PPARγ promotes alternative M2 macrophage activation (18, 19), we investigated macrophage polarization by evaluating the mannose receptor 1 (CD206, MRC1) expression. CD206 is a C-type lectin that is specifically expressed on the surface of M2 macrophages [18,26]. At 10 days, CD206 expression was undetectable in either sham- or MWCNT-instilled C57Bl/6 wild-type (Figure 1B). In contrast, CD206 was highly expressed in *Mmp12* KO sham- and MWCNT-instilled mice (Figure 1B). At 60 days, minimal expression of CD206 was observed and did not differ between sham- and MWCNT-instilled wild-type mice (Figure 1C). As expected, CD206 remained highly expressed at 60 days in both sham- and MWCNT-instilled *Mmp12* KO mice. Further studies were performed in *Mmp12* KO mice to explore the mechanism associated with granuloma resolution.

### 2.2. Mmp12 KO MWCNT Alveolar Macrophages Express M2c Markers at 10 Days and Convert to M2a by 60 Days

The M2 macrophages are further characterized into subtypes: M2a, M2b, and M2c. Surface expression of the receptor for advanced glycation end-products, or RAGE, is unique to M2c macrophages [27], while M2a macrophages are defined by Dectin-1 expression [28]. Currently, no unique surface proteins on M2b macrophages have been identified [28]. Macrophages from *Mmp12* KO mice were co-stained with anti-RAGE and CD206 to determine the presence of the M2c phenotype. Anti-Dectin-1 and CD206 were co-stained to delineate the M2a phenotype. As seen in Figure 2A, high levels of RAGE were observed in both *Mmp12* KO sham- and MWCNT-instilled mice at 10 days, while Dectin-1 was not observed in either group (Figure 2B). These results suggest that the M2c macrophage phenotype predominated at 10 days in both sham- and MWCNT-instilled *Mmp12* KO mice.

As granuloma resolution occurs at 60 days in *Mmp12* KO, we evaluated the macrophage phenotype using the same markers described above. At 60 days, RAGE expression was high in sham-instilled *Mmp12* KO but was absent in MWCNT-instilled *Mmp12* KO mice (Figure 3A). At 60 days, Dectin-1 was not expressed by sham-instilled *Mmp12* KO but was expressed at high levels in MWCNT-instilled *Mmp12* KO mice (Figure 3B). These observations of macrophage phenotype changes (summarized in Table 1) suggested a shift in macrophage polarization from M2c to M2a between 10 and 60 days in *Mmp12* KO mice instilled with MWCNT, coinciding with granuloma resolution.

### 2.3. IL-13 Gene and Protein Expression Are Elevated in BALF from Mmp12 KO Mice Instilled with MWCNT at 60 Days

IL-13 induces the M2a macrophage phenotype [29,30]. We investigated whether IL-13 was increased at 60 days. IL-13 gene and protein expression were significantly increased in MWCNT-instilled *Mmp12* KO at 60 days but not at 10 days (Figure 4A,B). Increased IL-13 was consistent with a phenotype shift from M2c to M2a in MWCNT-instilled *Mmp12* KO mice at 60 days, as seen in immunocytochemistry staining (Table 1). No change in IL-13 was observed in wild-type mice instilled with MWCNT (data not shown).

### 2.4. ApoE Is Upregulated in MWCNT-Instilled Mmp12 KO at 60 Days

PPARγ upregulation has been linked previously to upregulation of Apolipoprotein E (ApoE), a glycoprotein that is associated with numerous biological functions [31,32]. One of these functions involves the induction of the M2 macrophage phenotype [25,33,34]. Additionally, ApoE deficiency has been associated with sarcoid-like granulomatous formations [35]. We hypothesized that at 60 days, high levels of ApoE in *Mmp12* KO mice instilled with MWCNT might coincide with granuloma resolution.

Accordingly, ApoE gene expression was found to be elevated in MWCNT-instilled *Mmp12* KO at 60 days in comparison to 10 days (Figure 5A). ApoE gene expression was not elevated in MWCNT-instilled C57Bl/6 mice at 60 days (Figure 5A). Correspondingly, ApoE protein was increased in BALF from MWCNT-instilled *Mmp12* KO at 60 days relative to sham and 10-day ApoE levels (Figure 5B). No change in BALF ApoE protein expression was seen in C57Bl/6 mice (Figure 5B). At 10 days, low levels of ApoE protein were observed in BAL cells from both treatment groups of C57Bl/6 and *Mmp12* KO mice. However, increased ApoE expression was observed in BAL cells from 60-day *Mmp12* KO mice relative to C57Bl/6. These findings indicated that ApoE levels were low when granulomas were prevalent and increased when granulomas were resolving in 60-day MWCNT-instilled *Mmp12* KO mice.

### 2.5. Surface Expression of MMP14 Is Increased in the Absence of MMP12 at 60 Days

Recently, ApoE was implicated in the uptake and degradation of collagen fragments [36]. Atabai et al. demonstrated that collagen is degraded into fragments by surface proteins, including MMP14, and that these fragments are then bound by ApoE, marking the fragments for phagocytosis and intracellular degradation [37]. Collagen is a component of mature human sarcoid granulomas [6]. When immunocytochemistry was performed on BAL cells, an increase in MMP14 protein expression was seen at 60 days in MWCNT-instilled *Mmp12* KO mice relative to sham, while no expression was observed at 10 days (Figure 6). Thus, MMP14 protein expression coincides with BALF levels of ApoE, suggesting the activation of this pathway.

## 3. Discussion

As previously reported, granulomas resolved at 60 days in *Mmp12* KO mice while granuloma formation persisted in C57Bl/6 wild-type mice [17]. Thus, MMP12 is essential for the persistence of granulomatous formation through unknown mechanisms. In the MWCNT-instilled *Mmp12* KO, increased PPARγ expression coincided with this granuloma resolution [17]. PPARγ can function to change the phenotype of macrophages from M1 to M2 [19,20]. Functionally, this shift is from an inflammatory macrophage that acts by killing and clearing pathogens to an anti-inflammatory phenotype that acts in tissue repair [21]. As alveolar macrophages are the predominant immune cell in the lower airways where granulomas develop [8], we aimed to determine whether this phenotypic change could be involved in granuloma resolution. Our results using CD206, an M2 macrophage marker, in immunofluorescence staining clearly showed that the M2 phenotype was elevated in *Mmp12* KO mice with very low levels in C57Bl/6. Therefore, the M2 macrophage phenotype may be involved in granuloma resolution.

The M2 macrophage population can be further classified into M2a, M2b, and M2c subpopulations. We characterized the *Mmp12* KO macrophage population using M2a and M2c markers, Dectin-1 and RAGE, respectively [27,28]. Results from immunofluorescence analyses indicated the macrophage phenotype shifted from M2c to M2a between the acute (10 days) and chronic (60 days) stages of MWCNT-induced granulomas. The M2c macrophage phenotype was predominant during granuloma formation in MWCNT-instilled *Mmp12* KO mice, while the M2a phenotype was present during granuloma resolution. The M2a phenotype facilitates the release of M2-related cytokines, along with enhancing endocytic activity and promoting tissue repair [38]. This M2a function differs from that of the M2c, which functions in the phagocytosis of apoptotic cells [38]. The M2a phenotype and its corresponding functions may be responsible for the granuloma resolution at 60 days in *Mmp12* KO mice.

As a shift from M2c to M2a was observed in *Mmp12* KO mice, we attempted to determine whether IL-13 could be involved in this change. IL-13 is a cytokine that induces the M2a phenotype [30,31]. We observed increased levels of IL-13 in the *Mmp12* KO MWCNT-instilled mice at 60 days, which also corresponds with a shift to the M2a phenotype. At 10 days in these mice, granulomas formed in the presence of M2c macrophages. Between 10 and 60 days, the increased IL-13 resulted in a M2a macrophage phenotype when granulomas were resolving in MWCNT-instilled *Mmp12* KO mice.

PPARγ, which led us to explore macrophage polarization, has also been implicated in the regulation of ApoE [23,24]. Previously, Samokhin et al. showed that ApoE-deficient mice on a high cholate diet form sarcoid-like granulomatous formations [35]. Additionally, ApoE has also been shown to induce the M2 macrophage phenotype [25,33,34]. Correspondingly, ApoE expression is increased in MWCNT-instilled *Mmp12* KO mice at 60 days, when granulomas are resolving. The relationship between ApoE and granuloma resolution is not clear. However, ApoE was recently implicated in the uptake and degradation of collagen fragments [36]. ApoE binds these fragments, marking them for phagocytosis [37]. MMP14 is a surface protein that functions to degrade collagen [39]. We observed an increase in surface expression of MMP14 at 60 days in the *Mmp12* KO mice instilled with MWCNT. Taken together, these results led us to propose a model for granuloma resolution (Figure 7). At 60 days in MWCNT-instilled *Mmp12* KO mice, surface expression of MMP14 on the M2a macrophages degrades the collagen present in and around the granulomatous formation. Increased PPARγ expression leads to ApoE release by macrophages which bind collagen fragments. The bound fragments are then degraded by the M2a macrophages.

Even though, *Mmp12* KO mice instilled with MWCNT are intrinsically an M2 phenotype, our data suggest that the necessary M2c to M2a shift for granuloma resolution might be induced by IL-13. As MMP12 is necessary for granuloma persistence, our previous studies with a PPARγ agonist rosiglitazone have shown a decrease in granuloma size and formation in C57Bl/6 MWCNT-instilled mice [40], as well as downregulated MMP12 gene expression (Appendix A). These findings may have significant translational impact on granuloma resolution in sarcoidosis, as there is spontaneous remittance in 30% of the patients [1]. The exact pathogenesis of sarcoidosis is unknown. Similarly, why the disease in some patients resolves spontaneously is unknown. We are postulating that these spontaneous remissions may involve a shift in macrophage phenotypes and that potential therapies may involve agents which drive macrophages to an M2a phenotype. Future studies in patients are needed to better define macrophage phenotypes in patients with resolving disease versus patients with progressive disease. Interestingly, several studies have suggested that alveolar macrophages from sarcoidosis patients have a mixed phenotype with both M1 and M2 being present [41,42]. However, the M2 subtypes have not been investigated in sarcoidosis. In conclusion, future studies are needed to define macrophage subtypes, and involvement of matrix metalloproteinases along with ApoE, in patients with chronic non resolving granulomas and those whose disease has resolved.

## 4. Materials and Methods

### 4.1. Murine MWCNT Model

All studies were conducted in conformity with Public Health Service (PHS) policy on human care and use of laboratory animals and were approved by the institutional animal care and use committee (J207). C57Bl/6 wild-type and *Mmp12* KO mice (Jackson Laboratories, Bar Harbor, ME, USA) were administered a single oropharyngeal dose of freshly prepared MWCNT (100 µg) (900-1201, lot-GS1802, SES Research, Houston, TX, USA) in PBS/35% surfactant (gift: Ony Inc, Amherst, NY, USA) or PBS/35% surfactant alone (sham control) as previously described and characterized [43]. Animals were sacrificed at either 10 or 60 days post-instillation and evaluated as previously described [7].

### 4.2. Characterization of Bronchoalveolar Lavage (BAL) Cells

BAL cells were collected as previously described [7,44]. BAL cells were characterized by total cell counts and differential analysis (Table 2). Cells and aliquoted BAL fluid were stored at −80 °C and used for gene expression and protein analysis.

### 4.3. Immunocytochemistry

Cytospin slides of BAL cells were fixed with 4% paraformaldehyde–PBS. Following permeabilization (manufacturers’ instructions), cells were blocked with 10% goat serum–PBS. Slides were then stained with the corresponding primary antibody. Antibodies used were anti-CD206 at 1:100 (ab64693 Abcam, Cambridge, MA, USA), anti-Dectin-1 at 1:300 (ab140039 Abcam, Cambridge, MA, USA), anti-RAGE at 1:100 (ab37647 Abcam, Cambridge, MA, USA), anti-MMP14 at 1:1000 (ab78738 Abcam, Cambridge, MA, USA), and conjugated anti-CD206 488 at 1:200 (MCA2235A488T, Bio-Rad Laboratories, Hercules, CA, USA). Secondary antibodies were then added at 1:1000, using either Alexa 488 (A11011 Life Technologies, Carlsbad, CA, USA) or Alexa 568 (A11008 Life Technologies, Carlsbad, CA, USA). Slides were counter-stained with DAPI mounting medium (ab104139, Abcam, Cambridge, MA, USA) to facilitate nuclear localization. Following staining, the slides were imaged on the Zeiss LSM 700 Confocal (Zeiss, Oberkochen, Germany).

### 4.4. RNA Purification and Gene Expression from BAL Cells

Total RNA was extracted from frozen BAL cell pellets using miRNeasy Micro kit (217084, Qiagen, Germantown, MD, USA), according to manufacturer’s protocol. Mouse-specific primers were obtained from Qiagen, for ApoE (PPM04128B) and IL-13 (PPM03021B). GAPDH (PPM02946E) was used as a housekeeping gene. Quantitative-PCR was performed on complementary DNA synthesized using the RT2 First Strand Kit (330404, Qiagen, Germantown, MD, USA) and evaluated on the StepOnePlus PCR System (Thermo Fisher Scientific, Waltham, MA, USA) in comparison to GAPDH using the 2^−ΔΔCT^ method [45].

### 4.5. Protein Analysis of Bronchoalveolar Lavage Fluids (BALF)

ApoE (AB215086 Abcam, Cambridge, MA, USA) and IL-13 (M1300CB R&D Systems Inc., Minneapolis, MN, USA) were assayed in BALF by ELISA, per manufacturers’ protocol.

### 4.6. Statistical Analysis

Using Prism 7 software (GraphPad Inc., San Diego, CA, USA), the data were analyzed by Student’s t-test or one-way analysis of variance (ANOVA) with Tukey’s multiple comparison test. A *p*-value of ≤0.05 was considered significant.

### 4.7. Histological Analysis

Lungs were dissected and fixed in 10% buffered formalin. Paraffin embedded blocks were sectioned at 7 µm and stained with hematoxylin and eosin (H&E).

## Figures and Tables

**Figure 1 ijms-22-11019-f001:**
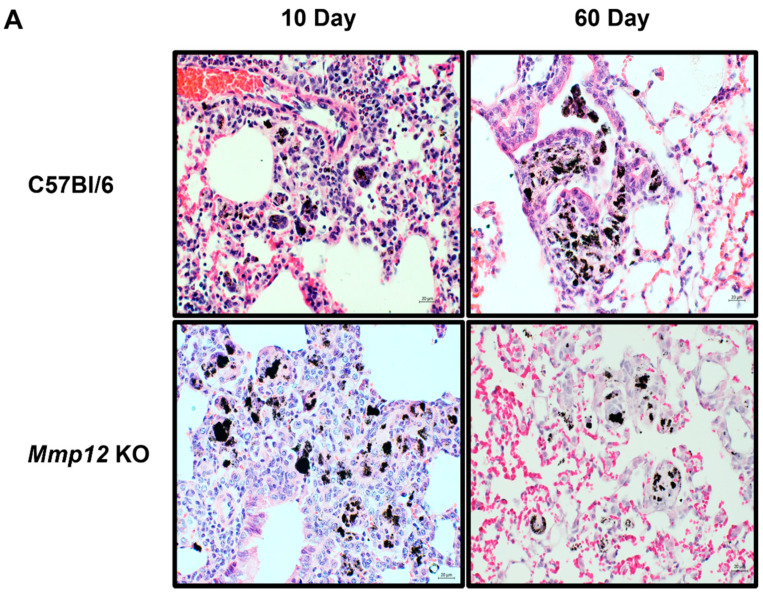
M2 is the predominant macrophage phenotype in *Mmp12* KO mice. (**A**) Lung tissue from MWCNT-instilled C57Bl/6 and *Mmp12* KO mice were stained with hematoxylin and eosin (H&E). (**B**) Immunofluorescent anti-CD206 staining of BAL cells showed an absence of CD206 protein in both the sham and MWCNT-instilled C57Bl/6 mice at 10 days. In *Mmp12* KO at 10 days, CD206 was highly expressed. (**C**) At 60 days, few cells stained positive in C57Bl/6 sham- and MWCNT-instilled mice. In the *Mmp12* KO strain, elevated expression of CD206 was observed in both MWCNT- and sham-instilled animals. Representative images of *n* ≥ 3 animals.

**Figure 2 ijms-22-11019-f002:**
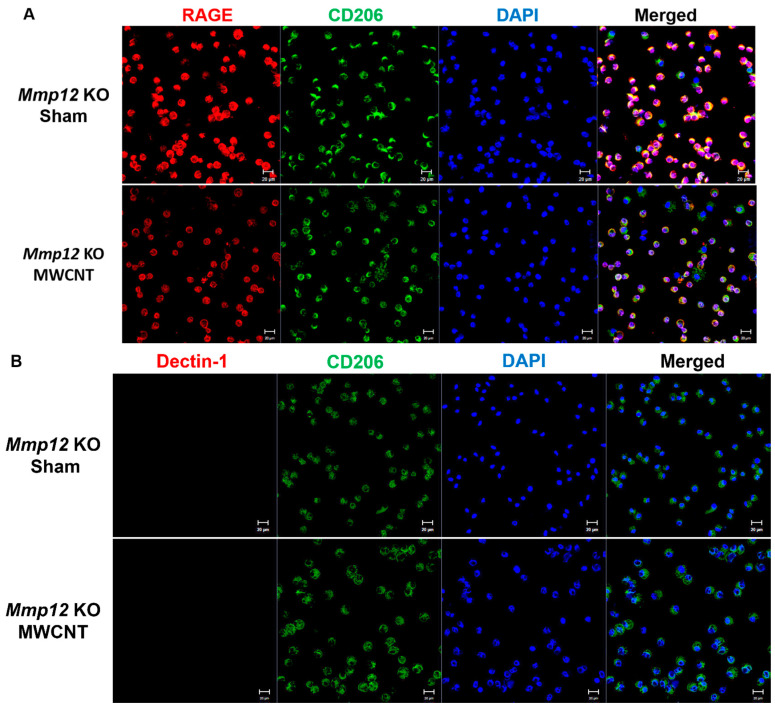
RAGE, a marker associated with the M2c macrophage phenotype, is prevalent in 10-day BAL cells from *Mmp12* KO mice. (**A**) Anti-RAGE staining of 10-day BAL cells from *Mmp12* KO MWCNT- and sham-instilled mice detected RAGE expression (co-stained with conjugated CD206). (**B**) Anti-Dectin-1 staining of *Mmp12* KO MWCNT and sham BAL cells showed no expression of the M2a macrophage marker, Dectin-1, at 10 days (co-stained with conjugated CD206). Representative images of *n* ≥ 3 animals.

**Figure 3 ijms-22-11019-f003:**
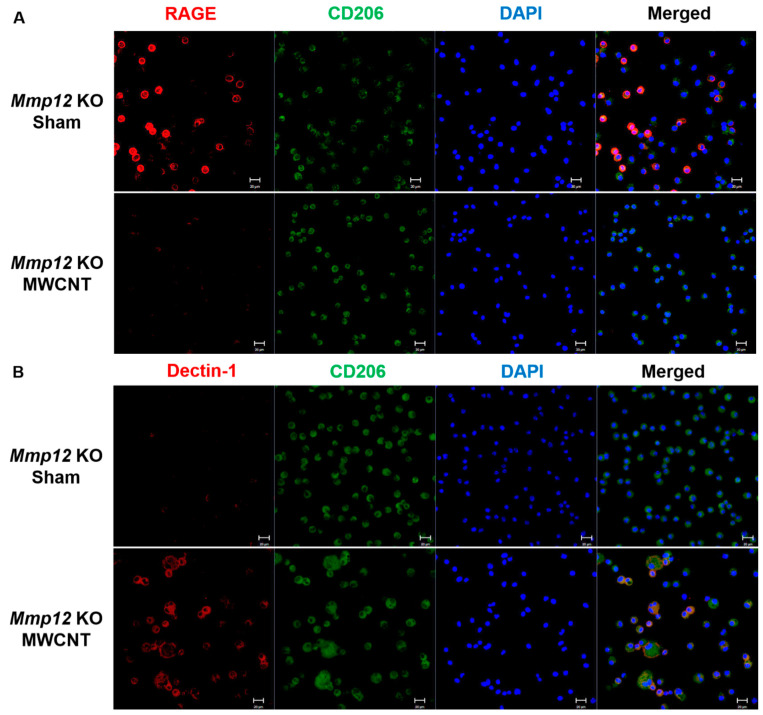
Dectin-1, an M2a macrophage phenotype, is prevalent in 60-day BAL cells from MWCNT-instilled *Mmp12* KO mice. (**A**) Immunofluorescent staining at 60 days showed high levels of RAGE expression in 60-day BAL cells from *Mmp12* KO sham, but low expression in MWCNT-instilled *Mmp12* KO mice. (**B**) Dectin-1 was not detected in BAL cells from *Mmp12* KO sham, but was highly expressed in BAL cells from MWCNT-instilled *Mmp12* KO mice. Representative images of *n* ≥ 3 animals.

**Figure 4 ijms-22-11019-f004:**
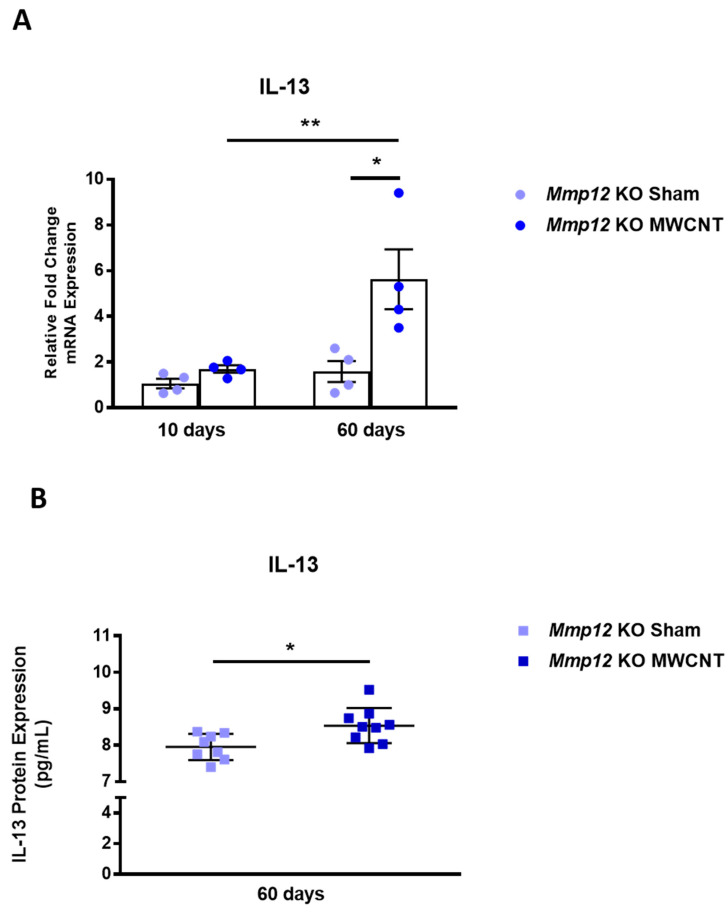
IL-13 gene and protein expression are increased in 60-day BAL cells from MWCNT-instilled *Mmp12* KO mice. (**A**) IL-13 gene expression was increased at 60 days, but no change was observed at 10 days. (**B**) IL-13 BALF protein levels were elevated in MWCNT-instilled *Mmp12* KO mice at 60 days relative to sham (** p* ≤ 0.05; ** *p* ≤ 0.01; *n* = 8/group). At 10 days, IL-13 protein levels were not detectable (data not shown).

**Figure 5 ijms-22-11019-f005:**
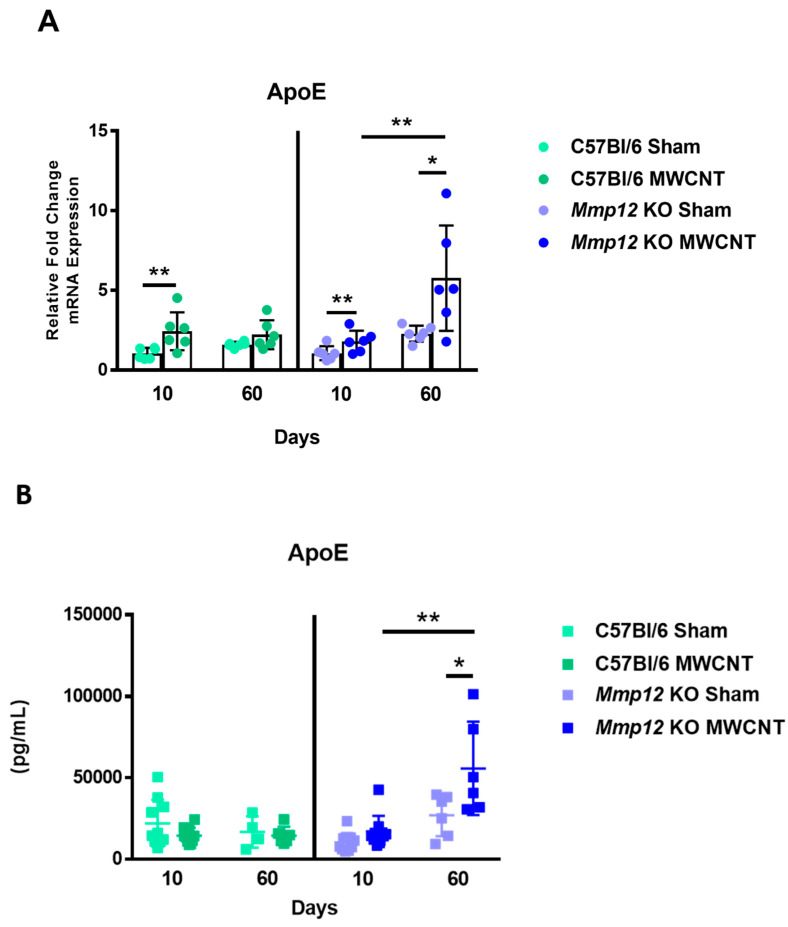
ApoE gene and protein levels are increased in MWCNT-instilled *Mmp*12 KO mice at 60 days. (**A**) ApoE gene expression was increased in MWCNT-instilled C57Bl/6 at 10 days, and in *Mmp*12 KO mice at both time points from BAL cells. ApoE was not increased in MWCNT-instilled C57Bl/6 mice relative to sham at 60 days. ApoE expression in MWCNT-instilled *Mmp*12 KO was increased at 60 days relative to 10 days (* *p* ≤ 0.05; ** *p* ≤ 0.01; *n* ≥ 5/group). (**B**) ApoE BALF protein levels were increased in MWCNT-instilled 60-day *Mmp*12 KO relative sham (* *p* ≤ 0.05; ** *p* ≤ 0.01; *n* ≥ 9/group).

**Figure 6 ijms-22-11019-f006:**
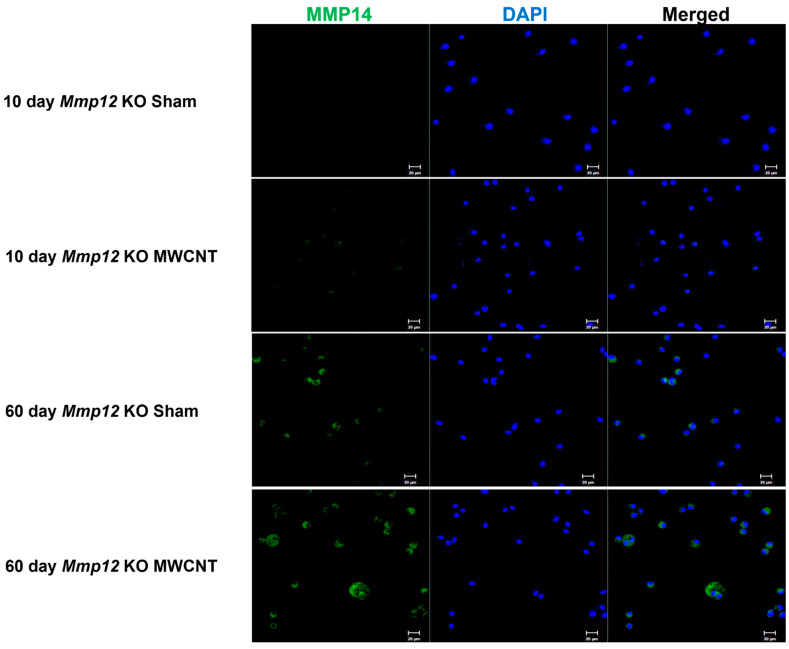
MMP14 protein increased in *Mmp12* KO mice at 60 days. ICC staining showed increased MMP14 expression on BAL cells from *Mmp12* KO mice at 60 days relative to 10 days. Additionally, *Mmp12* KO mice instilled with MWCNT expressed MMP14 at higher levels than in sham-instilled *Mmp12* KO mice. Representative images of *n* ≥ 3 animals.

**Figure 7 ijms-22-11019-f007:**
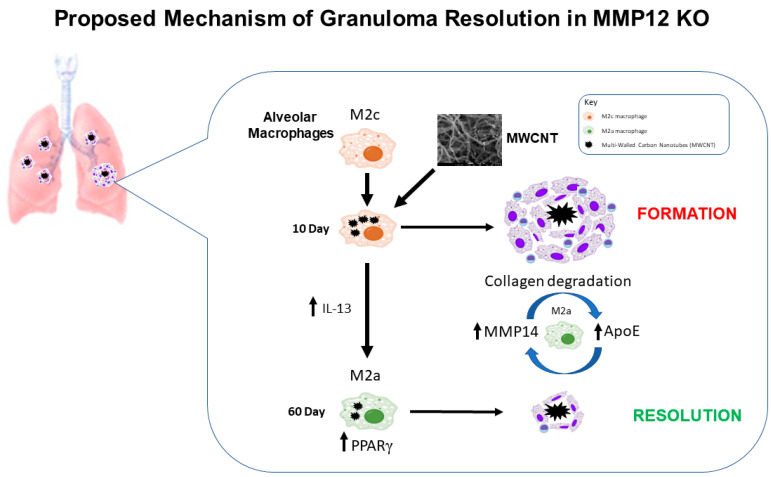
Schematic representation of the proposed mechanism. In *Mmp12* KO mice, M2c macrophages are the predominant immune cell. In the presence of MWCNT, granulomas form at 10 days in the *Mmp12* KO mice. An increase in IL-13 leads to the predominance of the M2a macrophage in MWCNT-instilled *Mmp12* KO mice at 60 days. This shift in M2 macrophage phenotype correlates with the resolution of granulomas at 60 days in *Mmp12* KO mice. We theorize that the role of M2a macrophages in granuloma resolution is through increased MMP14 surface expression which will function to degrade collagen, a defining feature of mature granulomas. These M2a cells also secrete increased ApoE, which can bind to collagen fragments, marking them for uptake. Through this collagen degradation and uptake, the stability of the granuloma will be diminished, leading to granuloma resolution.

**Table 1 ijms-22-11019-t001:** Summary of Macrophage Specific Markers.

*Mmp12* KO	CD 206 (M2)	RAGE (M2c)	Dectin-1 (M2a)
10 day Sham	+	+	−
10 day MWCNT	+	+	−
60 day Sham	+	+	−
60 day MWCNT	+	−	+

C57Bl/6 do not express M2 markers (data not shown); presence (+); absence (−).

**Table 2 ijms-22-11019-t002:** BAL cell characteristics of C57Bl/6 and *Mmp12* KO mice.

	Treatment	N	Total Cell Count (×10^5^)	AM (%)	LYM (%)	PMN (%)
**10 day**						
C57Bl/6	Sham	10	10.0 ± 4.3	99.3 ± 1.0	0.7 ± 0.9	0.0 ± 0.0
C57Bl/6	MWCNT	10	15.4 ± 4.7 *^a^*	87.7 ± 7.2	3.8 ± 2.5	8.5 ± 6.6
*Mmp12* KO	Sham	10	9.6 ± 1.4	98.9 ± 1.4	1.1 ± 1.4	0.0 ± 0.0
*Mmp12* KO	MWCNT	10	14.3 ± 6.0 *^a^*	93.4 ± 5.7	3.1 ± 2.1	3.5 ± 4.3
**60 day**						
C57Bl/6	Sham	9	6.6 ± 1.2	99.3 ± 1.0	0.7 ± 1.0	0.0 ± 0.0
C57Bl/6	MWCNT	10	7.1 ± 1.4	98.9 ± 1.3	0.9 ± 1.0	0.2 ± 0.6
*Mmp12* KO	Sham	10	10.1 ± 2.9	97.2 ± 4.7	1.3 ± 1.3	1.5 ± 3.8
*Mmp12* KO	MWNCT	10	17.7 ± 9.0 **^a^*	92.6 ± 4.8	5.3 ± 3.5	2.1 ± 1.7

Means ± SD; Alveolar macrophages (AM); lymphocytes (LYM); neutrophils (PMN); *^a^*
*p* ≤ 0.05 Sham vs. MWCNT; * *p* ≤ 0.05 *Mmp12* KO MWCNT vs. C57Bl/6 MWCNT.

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
