# Peer review of "The M2a Macrophage Phenotype Accompanies Pulmonary Granuloma Resolution in Mmp12 Knock-Out Mice Instilled with Multiwall Carbon Nanotubes"

_ijms, 2021, doi:10.3390/ijms222011019_

Round 1

Reviewer 1 Report

In this original research article, the authors want to demonstrate a newer macrophage phenotype under their previously published work in 2020-Knockout MMP12 that lung granuloma will be resolved at 60days after MWCNT installation. They observed a higher M2 macrophage in the BALF of MMP12 KO mice on 10 and 60 days. And they observed a phenotype transition in macrophage between Day 10 and 60 accompanied by an increased IL13. Meanwhile, the author also detected higher apoE and MMP14 on day 60. Thus, the author wants to illustrate that in MMP12 KO mice, macrophage transition led to a higher M2a cells, apoE, and MMP14 levels, which might help with the resolution of lung granuloma. 

I think this is very simple work and lack some logical evidence. The authors only used the samples from Day 10 and 60. No causal evidence is established here.

Here are some questions for the authors: 

  1. In figure 1, in MMP12 KO mice BALF cells, there is a relatively high level of M2 macrophages. What is the macrophage cell type in WT mice? Especially what kind of macrophage in WT sham group. The authors should add some other markers for these cells. 
  2. The author showed higher M2c macrophages in day10 and higher M2a macrophages on day 60, how could the authors address that knockout MMP12 will lead to the transition of these cells from M2c to M2a. Could you please add some in vitro experiments to show that- knockout MMP12 will lead to increased IL-13 secretion and phenotype change in these macrophages. 
  3. The authors suspect that apoE is a downstream target of PPARγ (as they talked in the introduction). Although authors showed an increased PPARγ after MMP12 KO in their previous work, that outcome needs to be established here along with apoE results. However, based on figure 5A, apoE is not correlated with PPARγ expression as MWCNT increases the apoE expression in C57BL6 mice but MWCNT lead to decreased PPARγ – this brings a big concern!

Other questions: 

  1. In figure 1A, the nuclear size of BALF cells were smaller MMP12 KO mice. Is that true or something wrong with the amplification. 
  2. In figure 5A, where is the sample come from? BALF cells or other tissue type?
  3. In figure 7, although an increased M2c cells in the BAL is observed, it shouldn’t be highlighted by an arrow because an increased M2 cells were due to MMP12KO instead of giving MWCNT. 

Author Response

Reviewer 1

Comment 1:  In figure 1, in MMP12 KO mice BALF cells, there is a relatively high level of M2 macrophages. What is the macrophage cell type in WT mice? Especially what kind of macrophage in WT sham group. The authors should add some other markers for these cells. 

Response 1:  Our previous studies have shown that WT mice instilled with MWCNT have a profile that is predominantly of the M1 phenotype.  Specifically, both alveolar macrophages from MWCNT-instilled WT mice and sarcoidosis patients express high levels of interferon gamma which is a hallmark for the M1 macrophages and sarcoidosis (Gharib et.al 2016, PMID: 27460362, Mohan et.al 2018, PMID: 29212802, Mohan et.al 2020, PMID: 33072094).  The WT sham group show very low gene and protein expression of interferon gamma (Mohan et.al 2020, PMID: 33072094).

Comment 2:  The author showed higher M2c macrophages in day10 and higher M2a macrophages on day 60, how could the authors address that knockout MMP12 will lead to the transition of these cells from M2c to M2a. Could you please add some in vitro experiments to show that- knockout MMP12 will lead to increased IL-13 secretion and phenotype change in these macrophages. 

Response 2:  We have tried to simulate the in vivo environment by exposing alveolar macrophages to MWCNT in vitro.  We were unable to achieve a cytokine response even with priming the macrophages with low dose LPS.  These experiments suggested in vitro conditions are not representative of the in vivo milieu. 

Comment 3:  The authors suspect that apoE is a downstream target of PPARγ (as they talked in the introduction). Although authors showed an increased PPARγ after MMP12 KO in their previous work, that outcome needs to be established here along with apoE results. However, based on figure 5A, apoE is not correlated with PPARγ expression as MWCNT increases the apoE expression in C57BL6 mice but MWCNT lead to decreased PPARγ – this brings a big concern!

Response 3:  C57BL6 mice at 60 days, (Fig 5A) show a slight increase in apoE mRNA expression, but this is not significantly different from sham instilled C57BL6 mice.  In addition, no increase in apoE protein was found (Fig 5B).  However, in MMP12 KO mice the gene and protein are increased at 60 days post MWCNT instillation as compared to sham.  At 60 days C57BL6 have low PPAR gamma expression while in the MMP12 KO PPAR gamma in high (see figure 7 in Mohan et.al 2020, PMID: 33072094).

Other questions: 

Comment 1:  In figure 1A, the nuclear size of BALF cells were smaller MMP12 KO mice. Is that true or something wrong with the amplification. 

Response 1:  The reviewer is correct in the observation that the cells are smaller. All the images are at the same magnification. 

Comment 2:  In figure 5A, where is the sample come from? BALF cells or other tissue type?

Response 2:  Figure 5A is BAL cells and 5B is BAL fluid

Comment 3:  In figure 7, although an increased M2c cells in the BAL is observed, it shouldn’t be highlighted by an arrow because an increased M2 cells were due to MMP12KO instead of giving MWCNT.

Response 3:  We have corrected the figure per the reviewer’s suggestion.

Reviewer 2 Report

Authors previously published their work which showed MMP12 is required for granuloma progression. Then,  in this stuy, they progressed their works using MMP12 KO mice and using MWCNT as the inducer or granuloma like sarcoidosis.   Then, they found that M2a macrophage phenotype accompanied pulmonary granuloma resolution in MMP12-KO mice instilled with MWCNT.    The text was well-written and all of the experimental presentation seemed to be well-performed.    The following issues would be altered by authors.   1) Even authors published previously, we hope to see the histological findings in 4 different groups.  2) If authors blocked development of M2a macrophage by some antibodies, such as anti-IL13 Ab, or MMP14 block, can we see not-resolved granuloma in some longer days such as 60 days after installation. 3) Please explain what was resolved in pathogenesis in sarcoidosis?  Because MMP-KO and installation of MWCNT were a kind of specific artificial situation, how we could use findings in this study for consideration of pathogenesis in sarcoidosis.  

Author Response

Reviewer 2

Comment 1. Even authors published previously, we hope to see the histological findings in 4 different groups. 

Response: We have added the histological findings to the manuscript (figure 1A).

Comment 2. If authors blocked development of M2a macrophage by some antibodies, such as anti-IL13 Ab, or MMP14 block, can we see not-resolved granuloma in some longer days such as 60 days after installation.

Response:  We agree with the reviewer.  Unfortunately, these experiments are not feasible in a timely manner because of the processes involved in getting animal use protocol approved by IACUC and then setting up animals with dose response curves and 60 day time points.

Comment 3:  Please explain what was resolved in pathogenesis in sarcoidosis?  Because MMP-KO and installation of MWCNT were a kind of specific artificial situation, how we could use findings in this study for consideration of pathogenesis in sarcoidosis.

Response 3:  We have added the following to the discussion to better explain the resolution in sarcoidosis“The exact pathogenesis of sarcoidosis is unknown.  Similarly, why the disease in some patients resolves spontaneously is unknown.  We are postulating that these spontaneous remissions may involve a shift in macrophage phenotypes and that potential therapies may involve agents which drive macrophages to an M2a phenotype.  Future studies in patients are needed to better define macrophage phenotypes in in patients with resolving disease versus patients with progressive disease.”  Furthermore, as the reviewer states our model is somewhat of an artificial situation.  However, manufactured carbon nanomaterials are used in building materials as well as many consumer products and maybe combustion generated during the burning of fossil fuels.  Interestingly, evidence suggesting potential involvement of carbon nanomaterial in human granulomatous disease has been gathered from analysis of dusts generated in the world trade center disaster combined with epidemiological data showing a subsequent increase in granulomatous disease of first responders (Wu M et.al, 2010 PMID: 20368128).  

Round 2

Reviewer 1 Report

NO more suggestion

Reviewer 2 Report

Authors modified their manuscript in this R1 version according to the reviewer's comments adequately.